# 1,3a,6a-Triazapentalene derivatives as photo-induced cytotoxic small fluorescent dyes

Daisuke Tsuji [1,2✉], Atsushi Nakayama[1,4], Riko Yamamoto[1], Shuji Nagano[1], Takashi Taniguchi[1], Ryota Sato[1], Sangita Karanjit[1], Naoki Muguruma[3,5], Tetsuji Takayama[3], Kohji Itoh[1] & Kosuke Namba [1✉]

1,3a,6a-Triazapentalene (TAP) is a compact fluorescent chromophore whose fluorescence properties vary greatly depending on the substituents on the TAP ring. This study investigated the photo-induced cytotoxicities of various TAP derivatives. Among the derivatives, 2-*p*-nitrophenyl-TAP showed significant cytotoxicity to HeLa cells under UV irradiation but no cytotoxicity without UV. In addition, the photo-induced cytotoxicity of 2-*p*-nitrophenyl-TAP was found to be cancer cell selective and effective against HeLa cells and HCT 116 cells. Under UV irradiation, 2-*p*-nitrophenyl-TAP generated reactive oxygen species (ROS) that induced an apoptosis and ferroptosis in cancer cells. Therefore, it was revealed that 2-*p*-nitrophenyl-TAP is the most compact dye that can generate ROS by photoirradiation.

[1] Graduate School of Pharmaceutical Sciences, Tokushima University, Tokushima 770-8505, Japan. [2] Department of Pharmacy, Faculty of Pharmacy, Yasuda Women's University, Hiroshima 731-0153, Japan. [3] Graduate School of Biomedical Sciences, Tokushima University, Tokushima 770-8505, Japan. [4] Present address: Graduate School of Science, Osaka Metropolitan University, 3-3-138, Sumiyoshi, Osaka 558-8585, Japan. [5] Present address: Takamatsu Municipal Hospital, Takamatsu, Kagawa 761-8538, Japan. ✉email: tsuji-d@yasuda-u.ac.jp; namba@tokushima-u.ac.jp

Chemotherapy is currently the most common treatment for cancer, but it always involves the serious problem of toxic side effects due to low selectivity for tumors. To reduce side effects, stimuli-responsive prodrugs that exhibit cytotoxicity in response to the specific stimulation in tumor tissue have been studied intensively[1–4]. Organic dyes that generate reactive oxygen species (ROS) by photoirradiation and cause cell damage have received a great deal of attention as promising stimuli-responsive prodrugs due to their high efficacy in killing cancer cells and their application to photodynamic therapy (PDT)[5–9]. So far, most of the dyes developed as ROS generators have been based on the molecular structures of existing fluorescent molecules such as porphyrins, phthalocyanines, chlorins, and fluoresceins[5–9]. To expand the possibilities of photoactive prodrugs and PDT, the development of structurally novel photoreactive dyes has always been required[10,11]. In particular, small molecular size dyes are expected to circulate efficiently in the body and are also expected not to affect the original behavior of the parent molecules when conjugated to target-affinity molecules such as antibodies, sugar chains, peptides, and proteins. So far, azulene has been reported as an example of a small dye for PDT, but there are still some points to be improved, such as weak cytotoxicity[12]. In 2011, we discovered that 1,3a,6a-triazapentalene skeleton (TAP) (1) is a compact and highly fluorescent chromophore (Fig. 1a)[13]. After revealing the fluorescence properties of TAP, we synthesized TAP analogs having various substituents[14–17] and expanded TAP to various fluorescent functional molecules such as fluorescent mechanochromic compounds[18], fluorescent-cell-staining reagents[19], fluorescent

bioactive functional groups[20], and iron ion sensors[21]. Other groups have reported alternative approaches for TAPs possessing different substituent patterns and their application to fluorescence bioimaging[22–28]. Recently, we developed a TAP analog, TAP-VK1, as a most compact fluorescent labelling reagent, and succeeded in visualizing the distribution of captopril, a small antihypertensive reagent, in vascular endothelial cells, demonstrating that the conjugation of the TAP ring did not affect the original biological activity of the parent molecule even though its molecular size was very small, such as that of captopril (MW: 217) (Fig. 1b)[29]. Therefore, we considered that if a TAP analog could generate ROS by photoirradiation, it would be an excellent photoreactive prodrug that could conjugate to molecularly targeted reagents. Throughout our previous studies on TAP, we found that its properties, such as its fluorescence intensity, fluorescence wavelength, chemical reactivity, photostability, and cytotoxicity, varied greatly depending on the substituents on the TAP ring. Therefore, considering that some of TAP derivatives might exhibit photo-induced cytotoxicity, we investigated their photoactivity using several TAP derivatives synthesized so far.

Herein, we report the successful development of an unconventional photo-induced cytotoxic small fluorescent dye based on TAP. The potent photo-induced cytotoxicity of the TAP derivative is derived from the efficient ROS generation, and the mechanism of cell-death is revealed to be ferroptosis at low concentrations and apoptosis at high concentrations. An unconventional ferroptosis inducer that can turn ferroptosis on by UV irradiation is developed in this study.

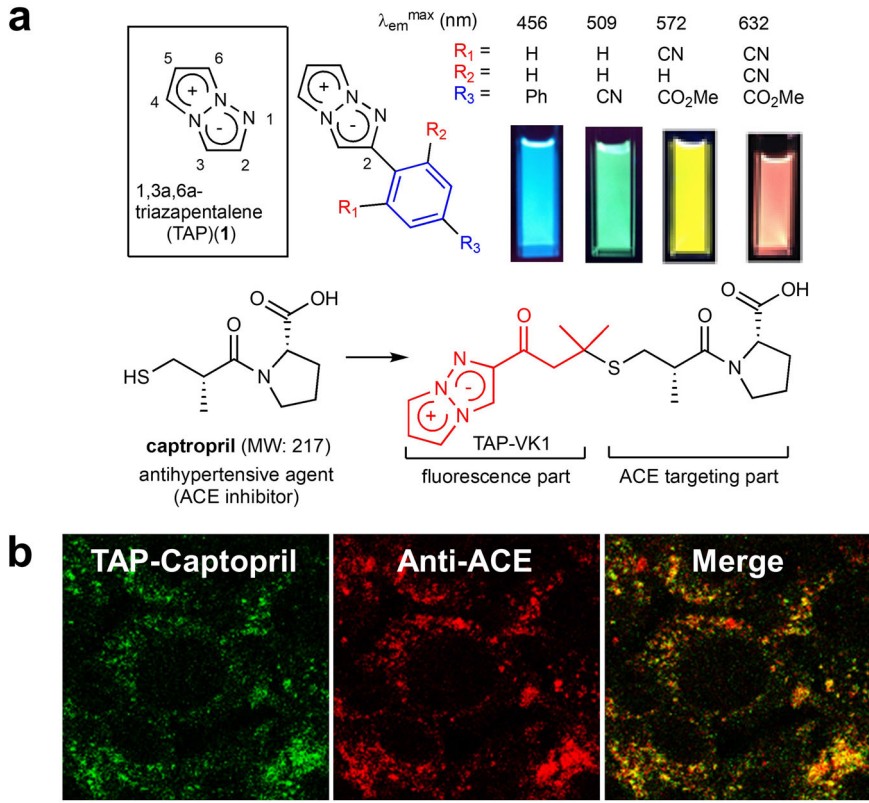

**Fig. 1 1,3a,6a-triazapentalene and its application. a** Structure of 1,3a,6a-triazapentalene and its fluorescence properties that change according to the inductive effect at the C2 position. **b** Development of TAP analog as a compact fluorescent labeling reagent and its application to the bioimaging of captopril, a small antihypertensive drug that inhibits angiotensin-converting enzyme. The captopril analog conjugated to TAP-VK1 showed fluorescence distribution consistent with that of antibody staining of ACE in vascular endothelial cells, while the conjugations of other fluorescent chromophores such as fluorescein, coumarin, nitrobenzoxadiazole (NBD), and aminosulfonylbenzoxadiazole (ABD), resulted in the disappearance of the original biological activity or showed a different fluorescence distribution to that of ACE. The TAP ring did not affect the original biological activity of the parent molecule even though its molecular size was very small, such as that of captopril.

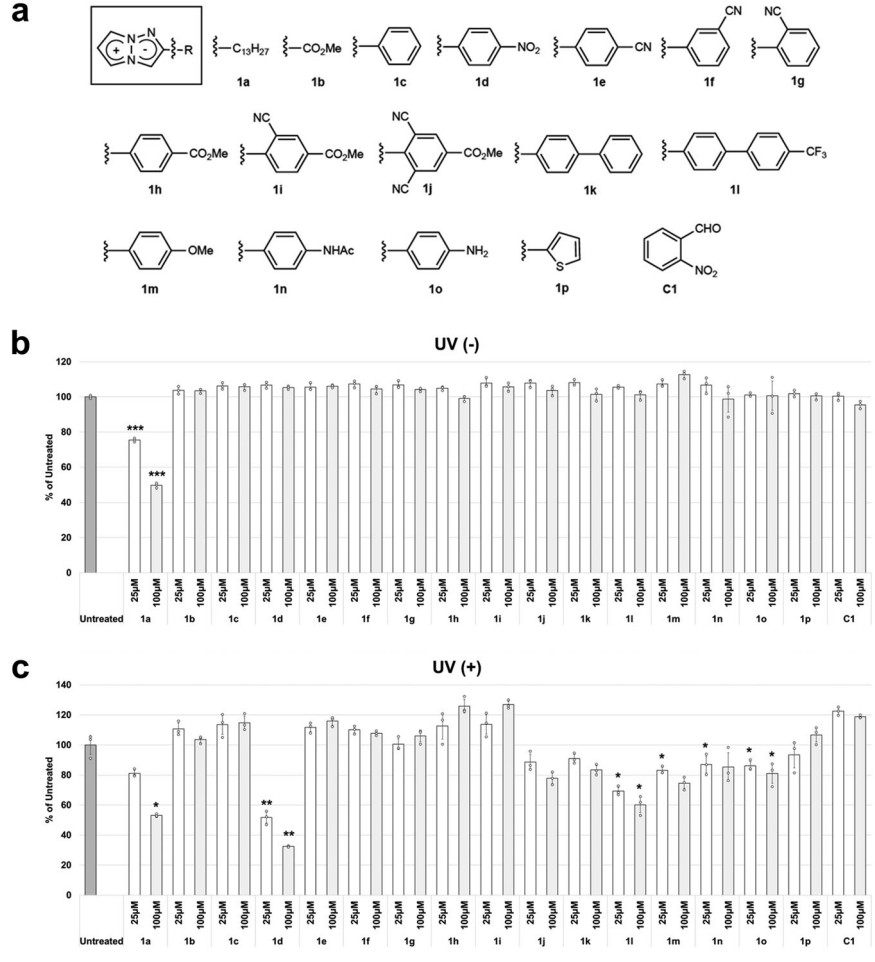

**Fig. 2 Screening for 2-substituted TAP derivatives exhibiting photo-induced cytotoxicity. a** Structures of TAP derivatives. **b** HeLa cells were cultured with 25 μM (black) or 100 μM (white) of each TAP derivative containing 0.1% DMSO as a control. After 1 h incubation, their cytotoxicities were analyzed by WST8 assay[34]. No cytotoxicity was observed without UV irradiation except for **1a**. **c** HeLa cells were cultured with 25 μM (black) or 100 μM (white) of each TAP derivative containing 0.1% DMSO as a control. After 1 h incubation, the cells were irradiated by UV (365 nm) for 1 h, and then their cytotoxicities were analyzed by WST8 assay. Several derivatives showed cytotoxicity under UV irradiation. Especially, *p*-nitrophenyl analog **1d** showed potent cytotoxicity in a concentration-dependent manner (**C1**: Negative control). *$P$ <0.05, **$P$ <0.01, ***$P$ <0.001 (vs Untreated) (Student's *t*-test) $N = 3$ biologically independent samples. The error bars represent the standard deviation of the mean.

## Results

**First screening**. The first screening of the photo-induced cyto-toxicity of TAP derivatives was performed using HeLa cells. Among the various substituent patterns of TAP derivatives, 2-substituted TAP derivatives were chosen in light of their availability, and the previously reported **1a-m** were examined (Fig. 2a) (Supplementary Table 1). Since the electron withdrawing groups at the 2-position of TAP increase chemical stability and fluorescence intensity, most of TAP derivatives synthesized so far have electron-withdrawing groups. To extensively investigate the photo-induced cytotoxicity, several TAP derivatives with electron-donating groups were also newly synthesized and evaluated (**1n-p**) (Fig. 2a). First, 25 and 100 μM solutions of TAP derivatives **1a-p** in culture medium (100 μL of DMEM 10% FBS) were treated with HeLa cells at 37 °C. The mixtures were incubated at 37 °C for 1 h, and then the cells were irradiated with ultraviolet (UV) (365 nm, 6 W) light from a distance of 5 cm for 1 h (Supplementary Fig. 1). Without UV irradiation, TAP derivatives showed no cytotoxicity in either the 25 μM or 100 μM solutions except for **1a**, suggesting that the alkyl chain of **1a** induced cytotoxicity and that the TAP ring is basically a non-cytotoxic chromophore (Fig. 2b). In contrast, with UV irradiation, the cell state varied significantly depending on the TAP

derivatives added. Several derivatives showed no significant cytotoxicity, while some others induced cell death; *p*-nitrophenyl analog **1d** showed a particularly potent effect in a concentration-dependent manner (Fig. 2c). On the other hand, only UV irradiation under the same condition (365 nm, 6 W, 5 cm distance, 1 h) without TAP analogs did not cause significant cell-death (Supplementary Fig. 2). Strong UV irradiation conditions usually damage cells, but the condition in this study did not show significant cytotoxicity probably due to the weak irradiation condition. In addition, blue LED irradiation also caused concentration-dependent photo-induced cytotoxicity of **1d**, albeit weaker than UV (Supplementary Fig. 3).

**Nitrophenyl-TAP analogs**. In light of the discovery of the photo-induced cytotoxic effect of *p*-nitrophenyl analog **1d**, we also synthesized *o*-nitro (**1q**), *m*-nitro (**1r**), and *o*, *p*-dinitrophenyl analogs (**1s**) from corresponding alkynes in a manner similar to the synthesis of **1d** to evaluate the substituent effect of the nitro group at different positions. As we expected, none of the derivatives showed cytotoxicity without photoirradiation (Fig. 3a). On the other hand, UV irradiation induced the cytotoxicity in all derivatives in a concentration-dependent manner, although the intensity of photoactivity varied according to the position of the

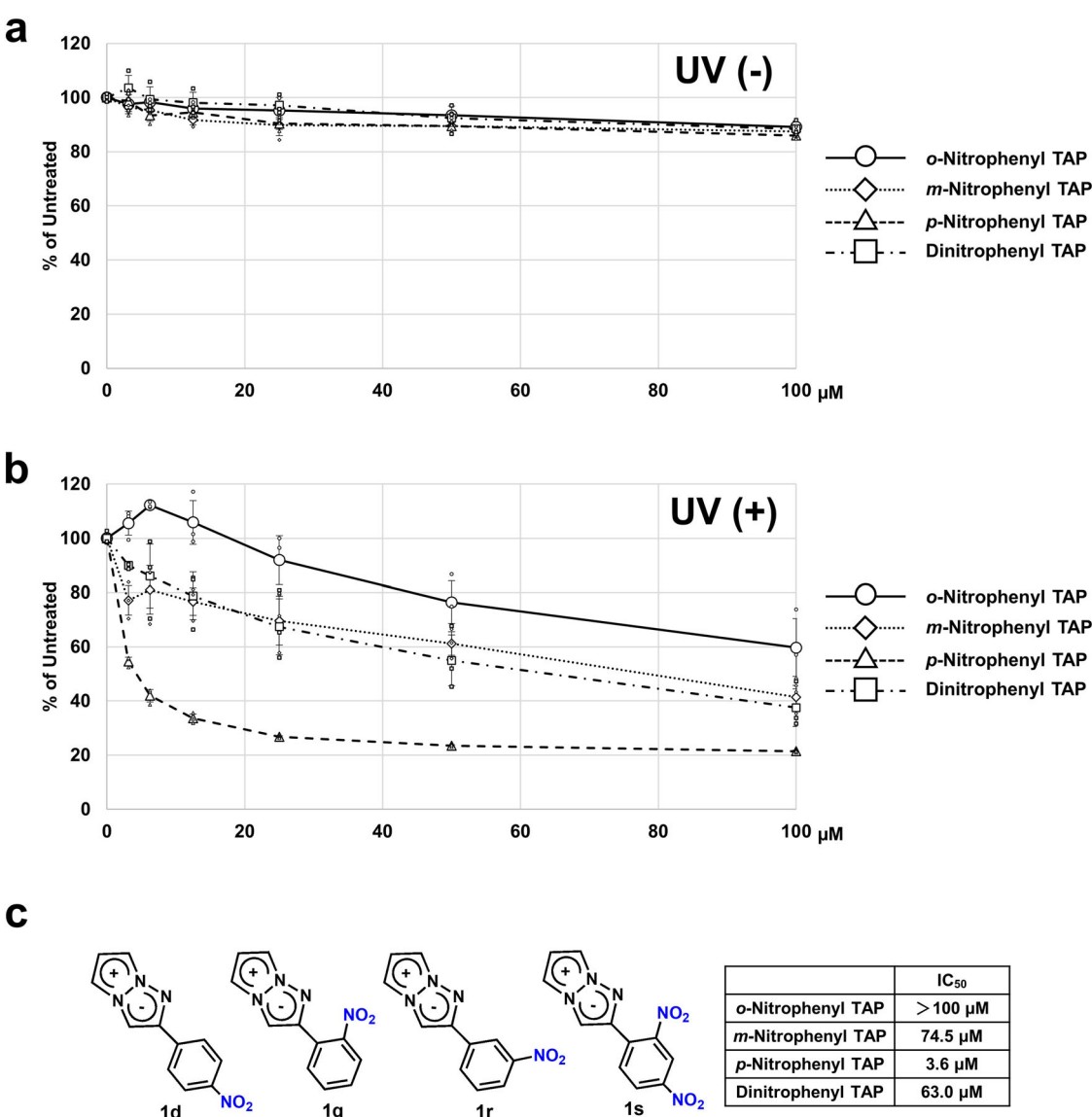

**Fig. 3 Photo-induced cytotoxic effect of nitrophenyl analogs. a** HeLa cells were cultured with several concentrations of TAP derivatives, and the cytotoxic effect without photoirradiation was determined by WST8 assay. **b** HeLa cells were cultured with several concentrations of TAP derivatives, and the photo-induced cytotoxic effect with UV irradiation was determined by WST8 assay. $N = 3$ biologically independent samples. The error bars represent the standard deviation of the mean. **c** Structures of nitrophenyl analogs **1d, 1q-s** and their IC$_{50}$ values. 4-Nitrophenyl analog **1d** showed much stronger photo-induced cytotoxicity than the other nitrophenyl analogs.

nitro group. Among nitro analogs, the *p*-nitro analog **1d** showed by far the strongest photo-induced cytotoxicity, with an IC$_{50}$ value of 3.6 μM, and the activities of the *m*-nitro analog **1r** and the *o,p*-dinitro analog **1s** were less potent than expected. In addition, the activity of *o*-nitro analog **1q** was considerably weak. Thus, the *p*-nitro analog **1d** was found to be the most promising so far (Fig. 3b, c).

**Cancer cell lines**. Next, the effectiveness of **1d** against other cancer cells was investigated. Interestingly, the photo-induced cytotoxicity of **1d** varied greatly depending on the type of cancer cells. While **1d** strongly induced the death of HeLa (cervical cancer) and HCT116 (colorectal cancer) cells by UV irradiation, weak cytotoxic activities were observed in RPMI8226 (myeloma), CCF-STTG1 (brain cancer), and A549 (lung cancer) cells, and there was no damage to GIST-T1 (gastrointestinal stromal tumor) cells (Table 1). Thus, the photo-induced cytotoxicity of **1d** was found to be selective for certain cancer cell lines, and the lines for

which **1d** was found effective can be applied to the endoscopic phototherapy.

**TAP uptake**. Having clarified the potential of **1d** to serve as an unconventional dye for PDT, we next investigated the mechanism underlying its cytotoxicity. First, to clarify the interaction of **1d** with HeLa cells, after 1 h incubation, the culture medium was

**Table. 1 IC$_{50}$ values of 1d in various cancer cell lines.**

|  | RPM18226 | CCF-STTG1 | GIST-T1 | A549 | HCT126 | HeLa |
|---|---|---|---|---|---|---|
| IC$_{50}$ (μM) | 63.2 | 76.1 | >100 | 93.2 | 8.6 | 3.6 |

The photo-induced cytotoxic activity of **1d** varied among cancer cell lines, and cell death was strongly induced in HeLa and HCT 116 cells.

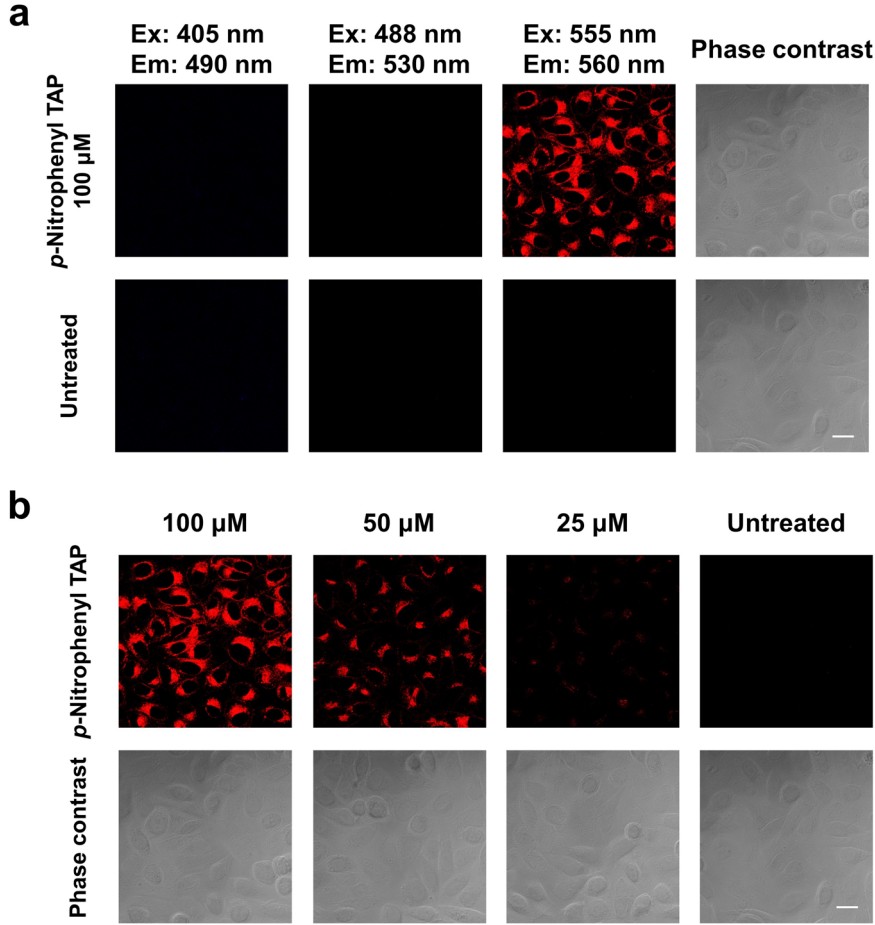

**Fig. 4 Observation of 1d in HeLa cells. a** HeLa cells were cultured with 100 μM of **1d** for 1 h, and then the cells were monitored by confocal microscopy (ZEISS LSM700; Carl Zeiss) in fluorescence images with various excitation and emission wavelengths. Since **1d** has no absorption band around 555 nm, it was considered that **1d** was converted to other substances inside the cells. **b** Living cells were cultured with 100, 50, and 25 μM of **1d** in fluorescence images with excitation and emission wavelengths of 555 and 560 nm, respectively. The fluorescence of **1d** was observed from inside the cells in a concentration-dependent manner, suggesting that **1d** was taken up by HeLa cells. Scale bar = 20 μm.

replaced and then UV irradiation was performed. **1d** again demonstrated photo-induced cytotoxicity, although the activity was slightly reduced compared to the case without replacement of the medium (Supplementary Fig. 4). Thus, **1d** was found to be taken up by cells or to interact with the cell surface. Similar activity was observed by UV irradiation without incubation after the treatment of **1d**, suggesting that the interaction or incorporation of **1d** was rapid (Supplementary Fig. 4). Then, we observed the fluorescence of the treated **1d** to clarify whether it was taken up by cells. After the application of various concentrations of **1d** solution and incubation for 1 h, the cells were washed and observed by a confocal laser microscope. Intracellular fluorescence was observed when the cells were excited at 555 nm, and the intensity depended on the concentration of **1d**, suggesting that **1d** was incorporated into cells (Fig. 4, and Supplementary Fig. 5 for magnified cell imaging). Interestingly, since **1d** has no absorption band around 555 nm, it was considered that **1d** was converted to other substances inside the cells. Indeed, when the culture medium was directly observed by the fluorescence microscope without washing, no fluorescence was observed from the culture medium even though a sufficient amount of unreacted **1d** was present; only the cells emitted fluorescence (Supplementary Fig. 6). Intracellular reduction of the nitro group of **1d** to the amino group by nitroreductase enzymes was considered as the possibility, but the synthesis of *p*-aminophenyl TAP **1o** and the evaluation of its fluorescence properties and photo-induced

cytotoxicity revealed that **1o** was not the active species (Fig. 2 and Supplementary Methods). We have no idea what **1d** was transformed into in the cells so far, but we plan to elucidate the structure of the metabolites in the future. The localization of **1d** was also investigated by co-staining with an organelle marker. The fluorescence of **1d** was observed as a dot shape in the cells. The fluorescence localization of **1d** overlapped mainly with the endosome marker Rab7 and also slightly matched with other organelles such as lysosome (see Supplementary Fig. 7). The quantification of the extent of localization by calculating the Pearson coefficients also suggested that **1d** localized mainly in late endosomes and was distributed slightly in most of the organelles involved in vesicle transport (early endosome, lysosomes, endoplasmic reticulum, Golgi apparatus, mitochondria, peroxisomes) (Supplementary Table 2).

**Intracellular fluorescence and cytotoxicity**. Having observed the fluorescence from inside HeLa cells, we investigated the relationship between fluorescence intensity and photo-induced cytotoxicity. The strongest fluorescence was observed from HeLa cells, and HCT116 cells also emitted stronger fluorescence than other cells. These were cells in which death was strongly induced by photoirradiation. On the other hand, the cells that exhibited weak photo-induced cytotoxicity did not show strong fluorescence except for CCF-STTG1. This suggested that the photo-induced cytotoxicity of **1d** was enhanced according to the

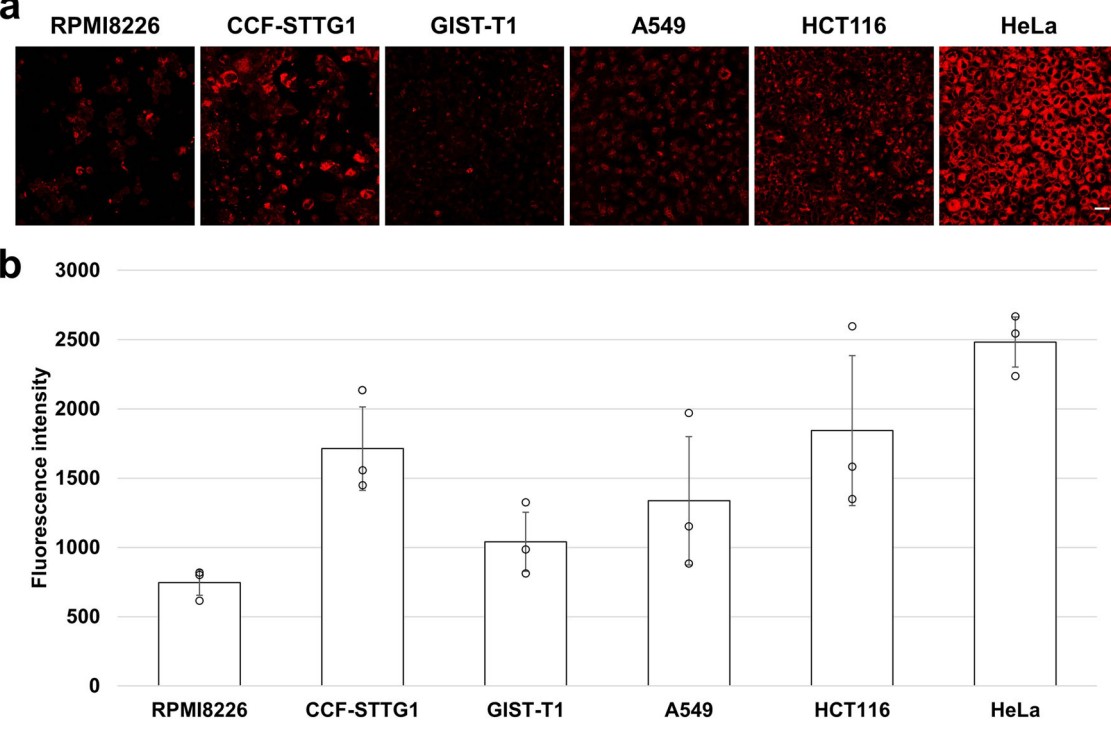

**Fig. 5 Fluorescence of 1d in various cancer cell lines. a** Living cells were cultured with 100 µM of **1d** for 1 h in fluorescence images with excitation and emission wavelengths of 555 and 560 nm, respectively. Scale bar = 50 µm. **b** Fluorescence intensity was measured by image analysis software (ZEN; Carl Zeiss) in each cancer cell line. The amount of **1d** uptake differed among the cell lines. Potent photo-induced cytotoxicity was observed in HeLa and HCT116 cells, and these cells took up more **1d** than the other cells. $N = 3$ biologically independent samples. The error bars represent the standard deviation of the mean.

amount of uptake into cells or the amount of conversion to the activated forms in cells (Fig. 5a, b, and Supplementary Fig. 8 for magnified cell imaging).

**ROS generation**. Next, the mechanism of cell death was investigated. While UV-induced cell death occurred in the concentration-dependent manner of **1d**, in the presence of *N*-acetylcysteine (NAC) as a radical scavenger, cytotoxicity was canceled regardless of the concentration even under UV irradiation (Fig. 6a, b). This result suggested the possibility that a reactive oxygen species (ROS) as an active species of cytotoxicity was generated from **1d** by UV irradiation and that the ROS was quenched by the radical scavenger. Thus, we used an ROS detection cell-based assay kit[30]. Without UV irradiation, no enhancement of fluorescence with the ROS detection kit was observed by treatment with 50 µM **1d**, and the fluorescence intensity was similar to untreated. Treatment with 500 µM $H_2O_2$ as a positive control significantly increased fluorescence with the detection kit ($P = 0.00429$ vs Untreated student's *t*-test), and the presence of 3 mM NAC suppressed the fluorescence enhancement (Fig. 6c). On the other hand, with UV irradiation, the fluorescence was increased even in the untreated cells, suggesting the ROS generation. Treatment with 50 µM **1d** significantly increased fluorescence to an intensity similar to that of $H_2O_2$ ($P = 0.00551$ vs Untreated student's *t*-test), and the addition of 3 mM NAC moderately suppressed the generation of ROS (Fig. 6c). Fluorescence observation by BIOREVO reflected the intensities of the graphs. Therefore, it was found that while **1d** does not generate ROS just by being taken up by cells, UV irradiation generated the same level of ROS as the treatment with $H_2O_2$. The comparison experiment using ROS Detection Cell-Based Assay Kit (DHE) revealed that the ROS generation efficiency of **1d** was similar to that of Rose Bengal, a well-known potent photosensitizer, and greater than that of Talaporfin sodium, a dye actually used for PDT in

clinical practice, indicating that **1d** exhibits high ROS generation efficiency (Supplementary Fig. 9).

**Mechanism of cell-death**. We next investigated whether cell-death was due to apoptosis by using apoptosis markers such as Annexin V and propidium iodide (PI) (Fig. 7a, b, and Supplementary Fig. 10 for magnified cell imaging). Fluorescence was not observed from Annexin V or from PI (nucleus) by an application of **1d** without UV irradiation, supporting the results of the cytotoxicity assay using WST-8 in Figs. 2–4. The weak cytoplasmic fluorescence observed at the wavelength for detecting PI (ex. 555 nm, em. 560) was the luminescence from the incorporated **1d**. Treatment with 500 µM $H_2O_2$ induced Annexin V-derived fluorescence at the cell membrane and did not induce PI-derived fluorescence, suggesting early apoptosis (Annexin V positive and PI negative). In addition, the fluorescence from Annexin V disappeared by the simultaneous application of NAC, and it was confirmed that the radical scavenger suppressed the apoptosis caused by $H_2O_2$. On the other hand, with UV irradiation, Annexin V-derived fluorescence was observed by the application of **1d** and PI-derived fluorescence was not observed. Also, no fluorescence was detected without treatment with **1d**. Thus, it was revealed that the combination of **1d** and UV induces early apoptosis as well as the treatment with $H_2O_2$. Moreover, the coexistence of NAC made Annexin V negative, suggesting that the generated ROS caused the apoptosis. Further detailed mechanisms are currently being investigated and so far have revealed that the nitro group of **1d** did not generate other cytotoxic species such as NO and $ONOO^-$[31,32].

**Ferroptosis**. Finally, we investigated the possibility that ferroptosis is involved in ROS-associated cell-death caused by **1d**. Ferroptosis, which is a type of programmed cell-death, is

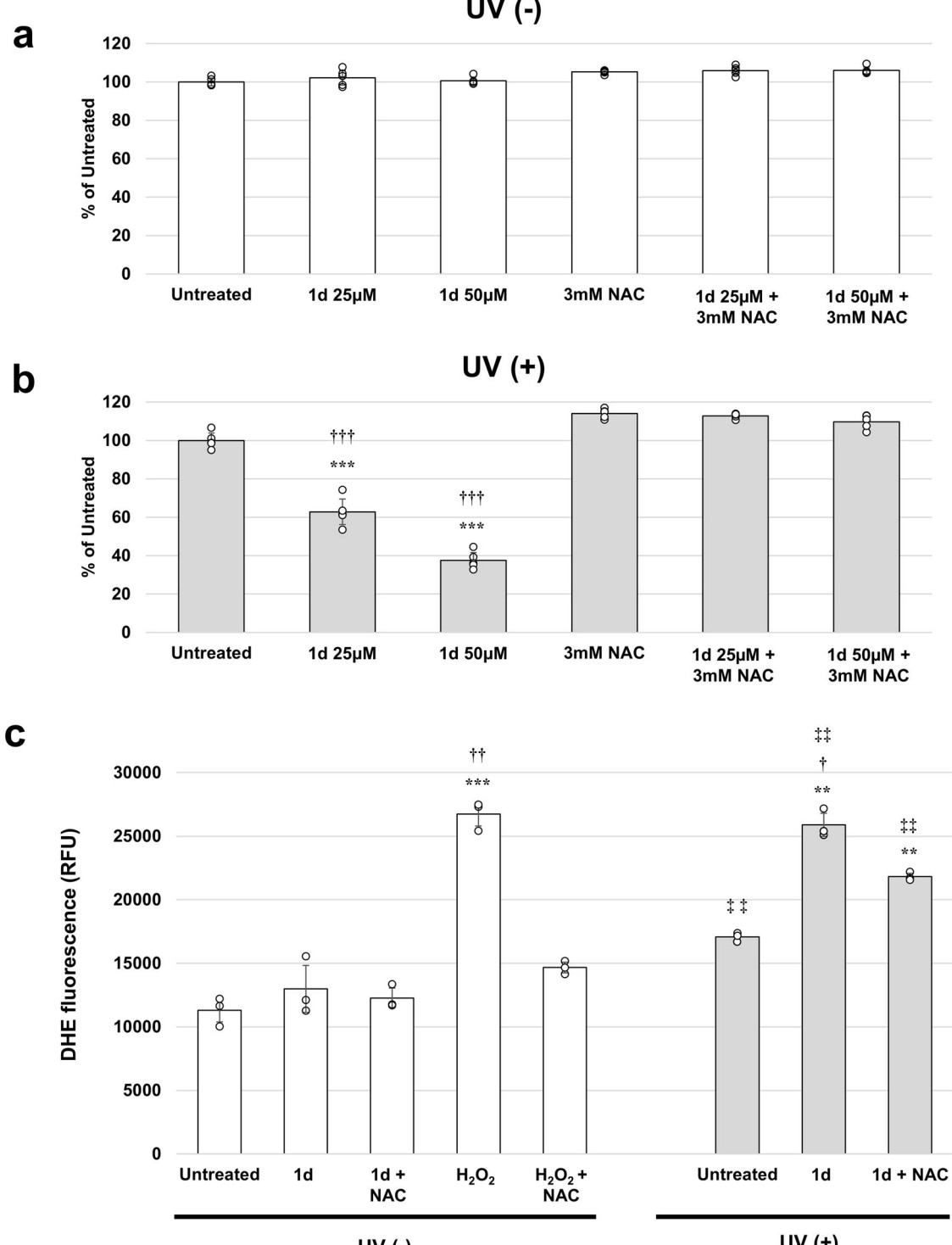

**Fig. 6 Production of reactive oxygen species (ROS) in 1d after UV irradiation. a** HeLa cells were cultured with 25 μM or 50 μM of **1d** for 1 h with or without 3 mM N-acetylcysteine (NAC), and then the cells were analyzed by WST8 assay without UV irradiation. **b** HeLa cells were cultured with 25 μM or 50 μM of **1d** for 1 h with or without 3 mM N-acetylcysteine (NAC), and then the cells were analyzed by WST8 assay with UV irradiation. **c** HeLa cells were cultured with 50 μM of **1d** or 500 μM $H_2O_2$ (as a positive control of ROS induction) for 1 h with or without 3 mM NAC, and then ROS in the cells was measured by a DHE probe (ROS Detection Cell-Based Assay Kit; Cayman). While **1d** did not generate ROS just by being taken up by cells, UV irradiation generated the same level of ROS as the treatment with $H_2O_2$. **$P$ <0.01,***$P$ <0.001 (vs Untreated), †$P$ <0.05, ††$P$ <0.01, †††$P$ <0.01 (vs with NAC), ‡‡$P$ <0.01 (vs UV (−)) (Student's $t$-test) $N = 3$ biologically independent samples. The error bars represent the standard deviation of the mean.

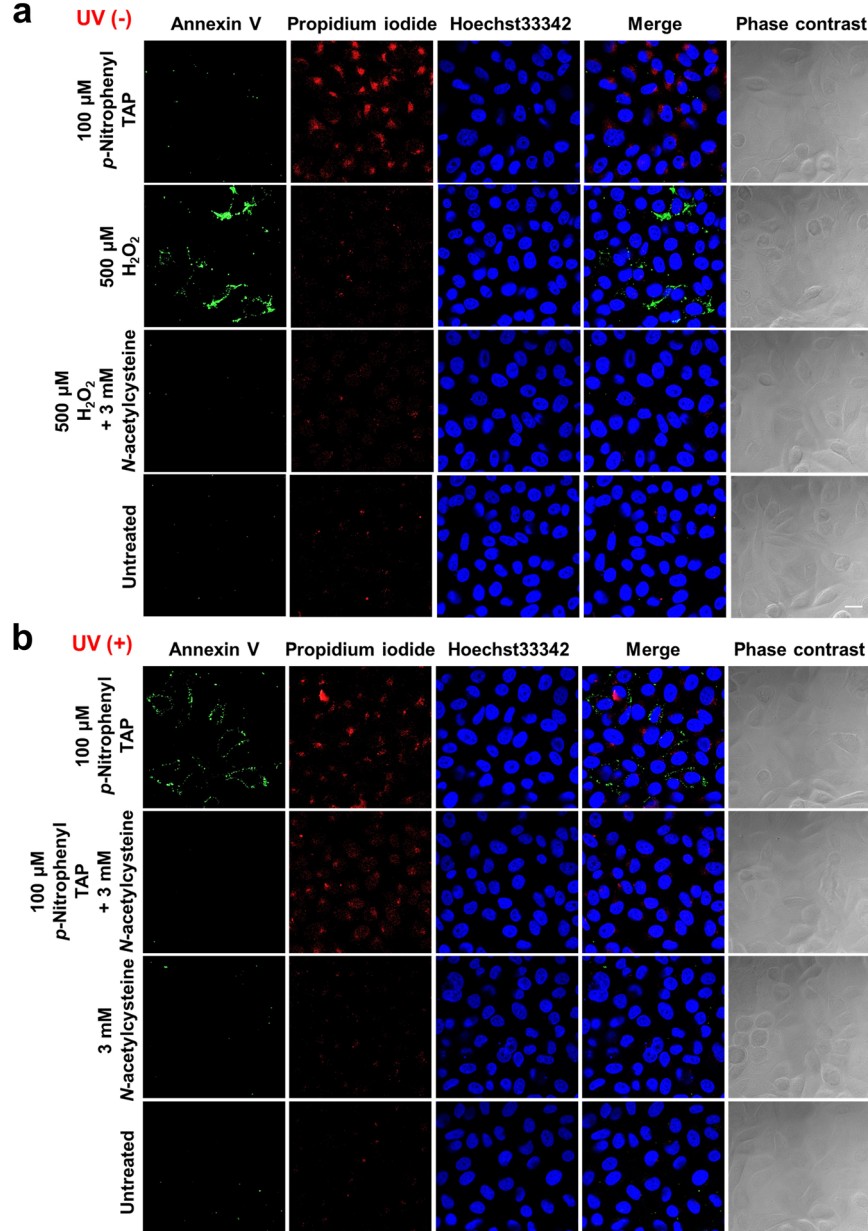

**Fig. 7 Detection of apoptosis in 1d after UV irradiation. a** HeLa cells were cultured with 100 μM of **1d** or 500 μM $H_2O_2$ for 1 h with or without 3 mM NAC, and the cells were incubated with FITC-Annexin V (Green), propidium iodide (PI) (Red), and Hoechst33342 (Blue), and observed by confocal microscopy. **b** HeLa cells were cultured with 100 μM of **1d** or 500 μM $H_2O_2$ for 1 h with or without 3 mM NAC, and treated with UV irradiation. After irradiation, the cells were incubated with FITC-Annexin V (Green), propidium iodide (PI) (Red), and Hoechst33342 (Blue), and observed by confocal microscopy. Scale bar = 20 μm.

characterized by a dysregulated iron metabolism and accumulation of lipid peroxides[33]. In the presence of Ferrostatin-1 as a ferroptosis inhibitor, UV-induced cytotoxicity of **1d** was canceled under UV irradiation in low concentrations (10 and 20 μM of **1d**). On the other hand, 50 μM of **1d** was not canceled by Ferrostatin-1 (Fig. 8a, b). Moreover, we detected labile iron in cells treated with **1d** by a fluorophore Mito-FerroGreen (Fig. 8c, and Supplementary Fig. 11 for magnified cell imaging). These results suggest that **1d** causes ferroptosis (low dose) and apoptosis (high dose) by UV irradiation. Since many ferroptosis inducers have not been reported so far, **1d** may be utilized as a inducer that can turn ferroptosis on by UV irradiation. To the best of our knowledge, this is the first example of the stimuli-responsive ferroptosis inducer.

## Conclusion

In conclusion, we have developed an unconventional photo-induced cytotoxic small fluorescent dye based on 1,3a,6a-triaza-pentalene (TAP). Among the various 2-substituted TAP derivatives, 2-*p*-nitrophenyl-TAP **1d** was found to exhibit potent photo-induced cytotoxicity against HeLa cells. The photo-induced cytotoxic activity of **1d** on cancer cells varied significantly depending on the type of cancer, and the activity was correlated with the amount of **1d** taken up into the cells. As the TAP skeleton is a fluorescent dye originally developed in our laboratory, 2-*p*-nitrophenyl-TAP (**1d**) is an unconventional ROS-generating dye for the photodynamic therapy (PDT). As the conventional dyes used for PDT have been limited mostly to relatively large molecules such as porphyrins, phthalocyanines, and fluorescein,

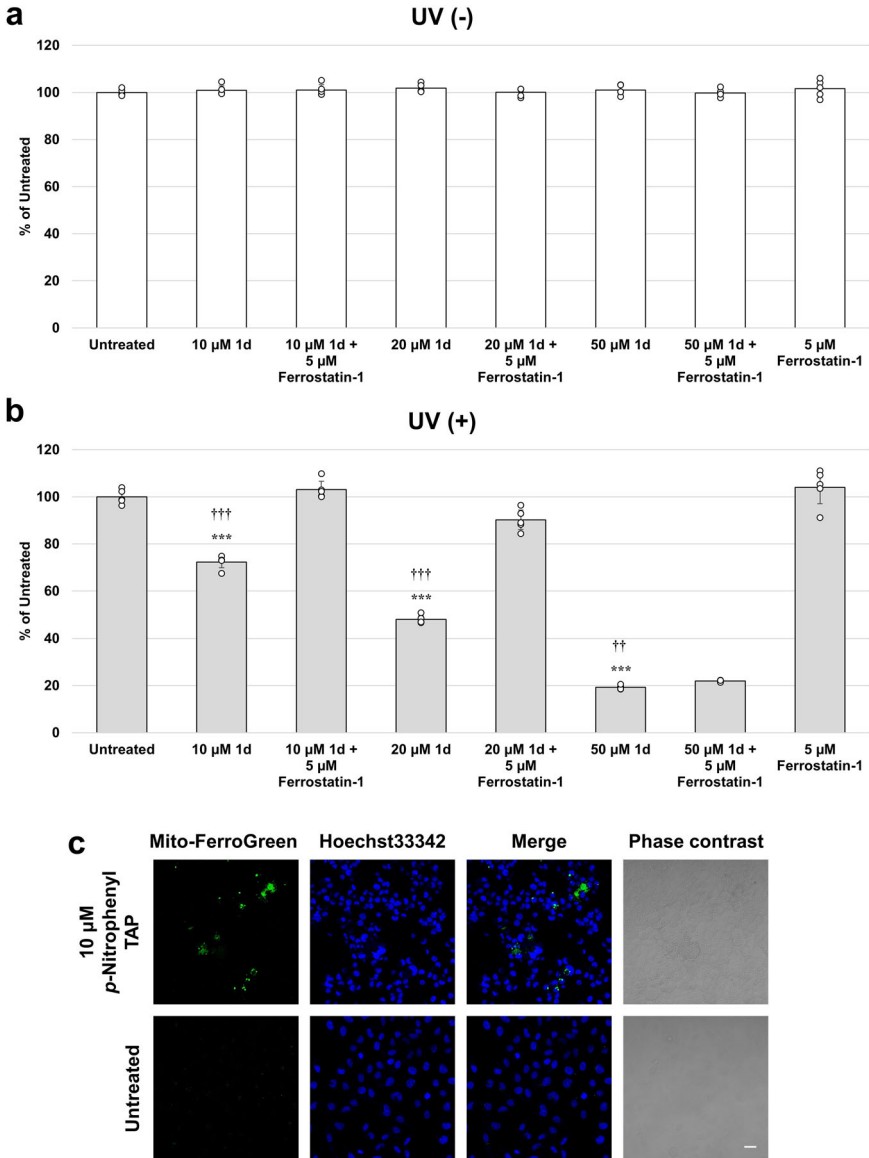

**Fig. 8 Detection of ferroptosis in 1d after UV irradiation. a** HeLa cells were cultured with 10, 20 µM and 50 µM of **1d** for 1 h with or without 5 µM Ferrostatin-1 (Ferroptosis inhibitor), and then the cells were analyzed by WST8 assay. **b** HeLa cells were cultured with 10, 20 µM and 50 µM of **1d** for 1 h with or without 5 µM Ferrostatin-1 (Ferroptosis inhibitor), and then the cells were analyzed by WST8 assay with UV irradiation. **c** HeLa cells were cultured with 10 µM of **1d** and 5 µM of Mito-FerroGreen for 1 h, and treated with UV irradiation. After irradiation, the cells were perfused and stained Hoechst33342 (Blue), and observed by confocal microscopy. Scale bar = 50 µm. ***$P$ <0.001 (vs Untreated),††$P$ <0.01, †††$P$ <0.01 (vs with Ferrostatin-1) $N$ = 5 biologically independent samples. The error bars represent the standard deviation of the mean.

they are difficult to introduce into a small target-affinity molecules such as drugs or natural products without losing their biological activities. The compact size of TAP analog is expected to support the further development of PDT. Although the use of UV as a light source is often problematic due to its limited penetration to tissue and cytotoxicity, cervical and colorectal cancers for which **1d** was found effective can be treated with direct irradiation from an endoscope. In addition, the UV irradiation condition in this study did not show significant cytotoxicity. Thus, applications of **1d** and its derivatives to in vivo experiments for colorectal, cervical, skin and oral cancers that can be directly irradiated with UV light are next subject. Also, to achieve the practical application of **1d** to PDT, further investigations for longer wavelength shifts and the conjugation of **1d** to target-affinity molecules are still required. It is known that substituents can be used to tune the wavelengths of TAPs, and we

have actually synthesized several derivatives with absorption maxima in visible light region. We are currently developing derivatives of **1d** with further longer wavelengths.

In this study, it was revealed that a TAP can serve as an ROS generating dye, revealing the potential of TAPs to serve as photoreactive prodrugs and for PDT. In addition, **1d** was found to induce ferroptosis in low concentration, and an unconventional ferroptosis inducer that can turn ferroptosis on by UV irradiation could be provided in this study. Further investigations toward a mechanistic elucidation of the photo-induced cytotoxicity of **1d** in details are currently under way in our laboratory.

## Methods

**Chemical synthesis**. Detailed synthetic procedures and compound characterization of novel TAP derivatives (**1n, 1o, 1p, 1q, 1r, and 1s**) are described in the Supplementary Methods, and [1]H and [13]C NMR spectra are available in

Supplementary Data 1. Absorption and fluorescence emission spectra are also available in Supplementary Data 2.

**Cell lines and culture**. The human cell lines, HeLa, A549, and HCT116 were provided by the RIKEN BRC through the National Bio-Resource Project of the MEXT/AMED, Japan. The human GIST cell line (GIST-T1) were kindly provided by Takahiro Taguchi (Kochi University, Kochi, Japan). The human multiple myeloma cell line (RPMI8226) was obtained from the American Type Culture Collection (Manassas, VA, USA). The human astrocytoma cell line (CCF-STTG1) was obtained from the DS Pharma Biomedical Co., Ltd (Osaka, Japan). Cells were grown in Dulbecco's Modified Eagle's Medium (DMEM: Sigma) containing 5% fetal bovine serum (FBS: Life Technologies), 70 mg/L penicillin (Sigma) and 100 mg/L streptomycin (Sigma), and maintained in the same culture medium at 37 °C in 5% $CO_2$.

**TAP solution in culture medium**. The TAP skeleton itself is lipophilic and the water-solubility of TAP analogs depend on the substituents. Since TAP derivatives examined in this study are lipophilic compounds, TAPs were applied as a culture medium-diluted dispersion. 40 μL of 10 mM TAP solutions in DMSO were diluted with 1 mL and 4 mL of culture medium (DMEM containing 5% FBS) to give 25 μM and 100 μM solutions, respectively.

**UV irradiation**. The prepared TAP solutions in culture medium (100 μL) were treated with cancer cell lines at 37 °C in 5% $CO_2$. The mixtures were incubated at 37 °C for 1 h, and then UV (6 W) was irradiated at a distance of 5 cm from the cells to the UV lamp (ASONE Handy UV Lamp SLUV-6) for 1 h.

**WST-8 assay**. Cancer cell lines ($1 \times 10^4$) were plated on collagen type I coated 96 well plate (IWAKI) and incubated with DMEM containing 5% FBS for a day. The medium was replaced with DMEM 5% containing compounds. After treatment of compounds and UV irradiation, the cells were treated with the Cell Counting Kit-8 (WST-8; Dojindo) according to the manufacturer's instructions. Briefly, the WST-8 reagent solution (10 μL) was added to each well of a 96 well microplate containing 100 μL of cells in the culture medium, and the plate was incubated for 2 h at 37 °C. Absorbance was measured at 450 nm using a microplate reader. Cell viability was expressed as a percentage of control (compound untreated with UV (+) or UV (−)). Cell survival rate (%) = (a-c) / (b-c) × 100 (a = absorbance at each concentration of compounds, b = absorbance at untreated with or without photo-irradiation, and c = absorbance of the blank). $IC_{50}$ values were calculated by linear approximation regression of the percentage survival versus the compound concentration.

**Confocal imaging**. Cancer cell lines ($5 \times 10^4$) were plated on 8 well chamber slide (Nunc) and incubated with DMEM containing 5% FBS for a day. The medium was replaced with DMEM 5% FBS containing compounds. The cells were fixed with 4% paraformaldehyde (PFA) after incubation for 24 h at 4 °C. Fixed cells were washed with phosphate buffered saline (PBS). The specimens were viewed under a confocal fluorescent microscopy (LSM700, Carl Zeiss).

**Measurement of cellular fluorescence intensity**. Digital imaging analysis was performed by a standard imaging software attached to the confocal microscopic system in a fluorescence mode (ZEN; Carl Zeiss). The cells containing significant fluorescent were designated as positive cells. Three different areas showing 20–30 cells each were viewed. The relative amount of fluorescence in the cell population was expressed as an intensity index in each image (mean value). Relative fluorescence intensity was graded, ranging from 0 to 255 in this system. The fluorescence intensity was linearly correlated with the number of pixels showing a positive signal. Relative intensities for other cell strains were calculated within the range of this gradation.

**Reactive oxygen species (ROS) assay**. Cancer cell lines ($2 \times 10^4$) were plated on collagen type I coated 96 well plate (IWAKI) and incubated with DMEM containing 5% FBS for a day. The medium was replaced with DMEM 5% FBS containing compounds or 500 μM $H_2O_2$ as a positive control. After treatment of compounds and UV irradiation, the cells were treated with the ROS detection cell-based assay kit (Cayman chemical) according to the manufacturer's instructions. Negative control was maintained by N-Acetyl cysteine to one group of untreated or treated cells. The cells were incubated for 1 h at 37 °C in dark, the ROS staining buffer was aspirated and 100 μl of DMEM 5% FBS was added to all the wells, the fluorescence was measured at 575 nm (Excitation: 488 nm). The ROS generated was expressed as total DHE fluorescence exhibited by the control and treated cells.

**Annexin V assay**. After treatment of compounds and UV irradiation, the HeLa cells were treated with FITC-Annexin V and propidium iodide solutions (Nacalai tesque) and 1 μg/mL Hoechst33342 (Sigma) for 1 h at room temperature. The cells were fixed with 4% PFA after incubation for 1 h at room temperature. Fixed cells were washed with phosphate-buffered saline (PBS), and then the specimens were viewed under confocal fluorescent microscopy.

**Mito-FerroGreen assay**. HeLa cells were treated with 5 μM Mito-FerroGreen (Dojindo) and 10 μM of **1d** for 1 h at 37 °C. After treatment of compounds and UV irradiation, the cells were fixed with 4% PFA after incubation for 1 h at room temperature. Fixed cells were stained Hoechst33342 (Sigma) and washed with phosphate-buffered saline (PBS), and then the specimens were viewed under confocal fluorescent microscopy.

**Measurement of absorbance/fluorescence spectra and fluorescence quantum yields**. Absorption spectra were recorded on a JASCO V-600 spectrometer and corrected fluorescence spectra were recorded on a JASCO FP-8200 spectrofluorometer. Sample solutions were degassed thoroughly by purging with an Ar gas stream for 30 min prior to the experiments and then sealed in their cells. All measurements were carried out at 25 °C. Fluorescence quantum yields were estimated by using 9,10-diphenylanthracene (9,10-DPA) in cyclohexane ($\Phi_F = 0.91$) or rhodamine B in ethanol ($\Phi_F = 0.94$) as a standard. Values were calculated by using the Eq. 1.

$$\Phi_x = \Phi_{st} \left[ A_{std}/A_x \right] \left[ F_x/F_{std} \right] \left[ n_x^2/n_{std}^2 \right] \tag{1}$$

std: standard, x: sample, A: absorbance at the excitation wavelength, F: fluorescence spectrum area, n: refractive index

**Statistics**. Group comparisons were performed using Student's t-test. Data are expressed as the mean ± standard deviation, and values were considered statistically significant at $P < 0.05$.

**Reporting summary**. Further information on research design is available in the Nature Portfolio Reporting Summary linked to this article.

## Data availability
The main data supporting the finding of this study are available within the paper and its Supplementary Information file. Other relevant data are available from the corresponding author upon reasonable request. NMR spectra of the compounds obtained in this manuscript are available in Supplementary Data 1. Absorption and fluorescence emission spectra are available in Supplementary Data. 2.

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

## Acknowledgements

The authors thank Dr. F. Yagishita (Institute of Post-LED Photonics, Tokushima University) for lending various LED lamps. This work was partially supported by JSPS KAKENHI Grant Number JP21K19051 and JP22H00352, and the Research Clusters program of Tokushima University (No. 1802001).

## Author contributions

K.N. conceived the chemical experiments and analyzed the results. D.T. conceived the biological experiments and analyzed the results. A.N., R.Y., S.N., T. Taniguchi, S.K. and R.S. performed the laboratory experiments and optimized the reaction conditions. N.M. and T. Takayama supplied cancer cells. K.N. and D.T. wrote the paper, and K.I. proofread the paper.

## Competing interests

The authors declare no competing interests.
