## [Peer Review File · Communications Chemistry]

Reviewers' comments:

Reviewer #1 (Remarks to the Author):

In this manuscript, the authors reported a series of triazapentalene derivatives and investigated their optical properties. The results showed that compound 2-p-nitrophenyl-TAP had significant cytotoxicity to HeLa cells under UV irradiation due to the generation of reactive oxygen species (ROS). This work presented some interesting results and almost all experiments supported the authors' claim. So this reviewer is of the opinion that this manuscript can be accepted to be published in Communications Chemistry only after major revision.

1) From the structures of triazapentalene derivatives, it is found that the benzene ring of all compounds contains electron-withdrawing groups. Are the electron-withdrawing groups very important for the generation of cytotoxicity under UV irradiation? Why not to try a compound containing electron donor?

2) For photosensitizers, excitation light is very important. However, these photosensitizers reported in this paper have excitation light located in the UV region. UV light itself is highly cytotoxic. How to go to exclude the interference of ultraviolet light? More, this will also greatly limit the application of these photosensitizers.

3) In all cell experiments, the cells are shown as one solid figure, which is completely different from the usual cell imaging. The authors need to give a reasonable explanation.

4) How about the cell cytotoxicity, water-solubility and fluorescent quantum yields of these compounds? The authors need to provide the related data.

Reviewer #2 (Remarks to the Author):

The authors tested the chemicals' potential as photosensitizer. However, the properties of the chemicals is questional for future practical applications.

1. The chemicals have absorption in UV wavelength region, but UV light can induce cell death and very limited penetration to tissue. How can the chemicals to be used in the future.

2. What is the ROS generation efficiency? Any comparison with well known photosensitizer?

3. One hour UV irradiation was applied to HeLa cells with different concentrations of chemicals incubated. It seems the data was processed by normalized to the 0 uM concentration cytotoxicity. It is not sure whether the UV irradiation can induce part of the cell death.

Reviewer #3 (Remarks to the Author):

Tsuji, Namba and co-workers have submitted a manuscript describing their evaluation of 1,3a,6a-Triazapentalenes (TAPs) as potential mediators of ROS ("reactive oxygen species") formation under photoirradiation. This work shows that this class of compound (and one compound in particular - 2-p-nitrophenyl-TAP) is indeed able to generate ROS in cellulo, leading to selective cell death under photoirradiation. In the longer term, this opens the possibility that TAP derivatives could be used for photodynamic therapy (PDT) to treat cancer.

The work has been performed proficiently and the manuscript is written and presented to a high standard. The principal claim to novelty is that TAP is a very small (i.e. low molecular weight) chromophore, and the authors argue that this is a significant advantage since such a small chromophore is less likely to perturb the biological function of any targeting agent to which it is attached. In support of this contention, they cite their own prior work on "TAP-VK1", in which

targeting of ACE by a derivative of captopril (a known ACE inhibitor) is successful despite this derivative having a TAP chromophore attached. Crucially, when other (larger) chromophores were used in analogous experiments, the selective recognition and labelling of ACE was not successful. Thus, the authors are claiming that what they have observed in a fluorescence labelling context (i.e. smaller chromophore leads to successful biological recognition event) will likely also be true in a PDT context. Ultimately I do find this argument convincing, and so I believe the criteria of novelty and significance for publication in *NatCommunChem* have been met.

I believe some minor revisions to the manuscript are required. Generally the references are appropriate and comprehensive. However, as the authors emphasise that low molecular weight should be advantageous in PDT, they should perhaps mention the use of azulene (another 10 π -electron bicyclic aromatic chromophore):

- Damrongrungruang et al., *Photodiagnosis and Photodynamic Therapy* **2018**, *24*, 318–323, <https://doi.org/10.1016/j.pdpdt.2018.10.015>

Other typos in the text should be fixed:

- Fig 1a: "lmaxem" should be " λ_{maxem} "
- "NBD", "ABD" – these acronyms should be defined.
- "UV (6 W)" – specify wavelength.
- Fig 3 caption: "c" should be "(c)"
- "mainly with" – extra space
- "stimuli-responsible" should be "stimuli-responsive"
- "N-Acetyl cysteine" should be "*N*-Acetyl cysteine"
- Ref 8: journal title italic.

Then, some minor scientific issues should be addressed:

- "Interestingly, since **1d** has no absorption band around 555 nm, it was considered that **1d** was converted to other substances inside the cells." – this is very significant, and the authors should write more about this. At the end of the paper the authors write that **1d** does not generate NO or ONOO⁻. But what substance(s) could **1d** be converted to? Is it possible that the nitro group is converted to an amine by nitroreductase enzymes? Is the *p*-amino-TAP already known? Does it absorb at 555nm? This would be easy to check.
- On the basis of Supplementary Figure 4, the authors state "The fluorescence localization of **1d** overlapped mainly with the endosome marker Rab7 and also slightly matched with other organelles such as lysosome (see supporting information)." This statement may be correct qualitatively, but it would be good to actually quantify the extent of localisation in the different cellular compartments by calculating the Pearson coefficients for each row in Supplementary Figure 4 and including these data in the supplementary information.

If the authors address all these issues I will recommend that the manuscript is accepted for publication.

Response to Reviewers:

Reviewer 1

1) From the structures of triazapentalene derivatives, it is found that the benzene ring of all compounds contains electron-withdrawing groups. Are the electron-withdrawing groups very important for the generation of cytotoxicity under UV irradiation? Why not to try a compound containing electron donor?

┘ Thank you for your important comments. Since the electron withdrawing group at the 2-position of 1,3a,6a-triazapentalene increase chemical stability and fluorescence intensity, most of TAP derivatives synthesized so far have electron-withdrawing groups. Therefore, we had many derivatives with electron-withdrawing groups and evaluated them. The evaluation of derivatives with electron donating groups is also important as you pointed out, so we synthesized thiophene analog in this study. To expand the scope of electron-donating groups according to your suggestion, we additionally synthesized and evaluate three derivatives (**1m**, **1o**, **1p**) in this revision. Also, the description of the effect of the electron-withdrawing groups at the 2-position was added to the main text.

2) For photosensitizers, excitation light is very important. However, these photosensitizers reported in this paper have excitation light located in the UV region. UV light itself is highly cytotoxic. How to go to exclude the interference of ultraviolet light? More, this will also greatly limit the application of these photosensitizers.

- Thank you for your important remarks. Although UV light is certainly cytotoxic, the photo-induced cytotoxicity of TAP analogs in this study was measured before the cell damage was induced by UV light. We added the experimental results that UV irradiation condition in this study (365 nm, 6W, 5cm distance, 1h) did not show significant cytotoxicity in the supplementary information (Supplementary Fig. 2). In the future, the TAP analog could be applicable to the treatment for skin cancer and oral cancer, and direct irradiation from an endoscope could be possible for cervical cancer and colorectal cancer for which efficacy were confirmed in this study. In addition, TAP analog **1d** was effective, albeit weaker than UV, even under blue LED irradiation, so it is also possible to develop the treatment using visible light. We added the results of changing light sources to the supplementary information (Supplementary Fig. 3). We added comments regarding the cytotoxicity of UV and the future applications of the TAP analog to the first screening section and conclusion section in the main text, respectively. The claim in this communication is that completely new and compact dye that exhibits photo-induced cytotoxicity was discovered by the application expansion of TAP skeleton, and it will be the first reagent that induces ferroptosis by photo-stimulation. Thus, we think that the study on practical application is the next issue after this communication. As the fluorescence properties of TAP analogs varied widely with the 2-substituents, the modification of the nitrophenyl moiety may allow a longer wavelength shifts of the absorbance maximum. Actually, we already synthesized some nitrophenyl analogs that exhibit stronger photo-induced cytotoxicity by blue LED, and studies on further longer wavelength shift is currently underway in our laboratory. The development of another analog that can be excited by green or red LED will be reported as a full paper in the future.

3) In all cell experiments, the cells are shown as one solid figure, which is completely different from the usual cell imaging. The authors need to give a reasonable explanation.

- Thank you for your advice. We added magnified cell imaging to supplementary information (Supplementary Figs. 5, 8, 10, and 11). If we do not understand your point correctly, we would appreciate it if you let us know.

4) How about the cell cytotoxicity, water-solubility and fluorescent quantum yields of these compounds? The authors need to provide the related data.

- Thank you for the important comments. As shown in Figure 2, most TAPs did not show the cell cytotoxicity without UV irradiation, indicating that TAP skeleton itself is not cytotoxic. To readily find the cytotoxicity of TAPs, the experimental procedure for the cell cytotoxicity was described in more detail in the Methods section. The TAP skeleton itself

is lipophilic and the water-solubility of TAP analogs depend on the substituents. Since TAP analogs examined in this

study are lipophilic compounds, TAPs were dissolved in DMSO and applied as a culture medium-diluted dispersion. We added the preparative method of culture medium solution of TAPs to Methods section, and the water-solubility of TAP derivatives was mentioned. The method for calculating the fluorescence quantum yield was also added to Method section. Most TAP analogs in Figure 1 were already reported by our group, so their fluorescence properties including fluorescence quantum yields were summarized in Supplementary Table 1 with their reference. Full data of new TAP analogs **1m-s** synthesized in this study were described in Supplementary data, and their ¹H and ¹³C NMR charts and fluorescent spectra were also attached.

Reviewer 2

1) The chemicals have absorption in UV wavelength region, but UV light can induce cell death and very limited penetration to tissue. How can the chemicals be used in the future.

- Thank you for your important remarks. Since the UV irradiation condition (365 nm, 5 cm distance, 6W for 1h) in this study did not induce significant cell death, it is possible to directly irradiate cancer cells with UV under the same condition. We mentioned about no cytotoxicity of only UV irradiation in first screening section in the main text and added the experimental result of only UV irradiation to Supplementary information (Supplementary Fig. 2). In particular, cervical cancer and colorectal cancer, for which **1d** was selectively effective, can be directly irradiated with light from an endoscope, so we will target these cancers for the time being. The brief description of this idea for future use was added to the conclusion in the main text. In addition, TAP analog **1d** was effective, albeit weaker than UV, under blue LED irradiation, so it is also possible to develop the treatment using visible light. We added the results of changing light source to the supplementary information (Supplementary Fig. 3). Furthermore, as the fluorescence properties of TAP analogs varied widely with the 2-substituents, the modification of the nitrophenyl moiety may allow a longer wavelength shifts of the absorbance maximum. Indeed, we already synthesized some nitrophenyl analogs that exhibit stronger photo-induced cytotoxicity by blue LED, and studies on further longer wavelength shift is currently underway in our laboratory. However, the claim in this communication is that completely new and compact dye that exhibits photo-induced cytotoxicity was discovered by the application expansion of TAP skeleton, and it will be the first reagent that induces ferroptosis by photo-stimulation. Thus, we think that the study on practical application is the next issue after this communication.

0) What is the ROS generation efficiency? Any comparison with well known photosensitizer?

- Thank you for making the important point. To evaluate the ROS generation efficiency of **1d**, the additional experiment was conducted to compare with Rose Bengal which is a well-

known potent photosensitizer and Talaporfin sodium which is actually used for a photodynamic therapy

in clinical practice by using ROS Detection Cell-Based Assay Kit (DHE). The ROS generation efficiency of **1d** was found to be similar to that of Rose Bengal and greater than that of Talaporfin sodium, indicating that **1d** exhibits high ROS generation efficiency. This experimental result was added to Supplementary information (Supplementary Fig. 9) and the brief description about ROS generation efficiency of **1d** was also added to the main text.

3) One hour UV irradiation was applied to Hela cells with different concentrations of chemicals incubated. It seems the data was processed by normalized to the 0 uM concentration cyto viability. It is not sure whether the UV irradiation can induce part of the cell death.

- Thank you for the important point. The data was actually processed by normalized to 0 μ M concentration cytoviability, but the UV irradiation condition in this study (365nm, 6W, 5 cm distance, 1h) did not induce significant cell death. Therefore, the data is almost equivalent to the non-normalized one. Evaluation of cell damage when only UV was irradiated under same conditions was added to the Supplementary data (Supplementary Fig. 2). Also, the description that only UV irradiation did not induce significant cell-death was added to the main text.

Reviewer 3

I believe some minor revisions to the manuscript are required. Generally the references are appropriate and comprehensive. However, as the authors emphasise that low molecular weight should be advantageous in PDT, they should perhaps mention the use of azulene (another 10 π -electron bicyclic aromatic chromophore):

• Damrongrungruang et al., *Photodiagnosis and Photodynamic Therapy* **2018**, *24*, 318-323, <https://doi.org/10.1016/j.pdpdt.2018.10.015>

- Thank you for your valuable suggestion. We mentioned azulene as a compact photosensitizer for PDT in the main text and cited the suggested reference.

Then, some minor scientific issues should be addressed:

• "Interestingly, since **1d** has no absorption band around 555 nm, it was considered that **1d** was converted to other substances inside the cells." – this is very significant, and the authors should write more about this. At the end of the paper the authors write that **1d** does not generate NO or ONOO⁻. " But what substance(s) could **1d** be converted to? Is it possible that the nitro group is converted to an amine by nitroreductase enzymes? Is the *p*-amino-TAP already known? Does it absorb at 555nm? This would be easy to check.

- Thank you for making the important point. Although it took a long time to synthesize, we succeeded in the synthesis of 2-(*p*-aminophenyl)-TAP by using the Teoc group as a protecting group for the amino group after various examinations. The absorption

spectrum of 2-(p-aminophenyl)-TAP showed no band around 555 nm, and no photo-induced cytotoxicity was also

observed. Thus, it found that 2-(p-aminophenyl)-TAP was not the true active species. Although the reduction of the nitro group of **1d** to the amino group was not the actual transformation occurring intracellularly, this was important consideration. We appreciate your adequate suggestion. Description about 2-(p-aminophenyl)-TAP was added to the main text, and the synthesis and data of 2-(p-aminophenyl)-TAP were described in the Supplementary data. So far, we have no idea what **1d** was transformed into in the cells, but we plan to elucidate the structure of the metabolites and report them in the future.

- On the basis of Supplementary Figure 4, the authors state “The fluorescence localization of 1d overlapped mainly with the endosome marker Rab7 and also slightly matched with other organelles such as lysosome (see supporting information).” This statement may be correct qualitatively, but it would be good to actually quantify the extent of localisation in the different cellular compartments by calculating the Pearson coefficients for each row in Supplementary Figure 4 and including these data in the supplementary information.

- Thank you for the good suggestion. As following your suggestion, we quantify the extent of localization in the different cellular compartments by calculating the Pearson coefficients, and added them to Supplementary Information (Supplementary Table 2).

Other typos in the text should be fixed:

- Fig 1a: “lmaxem” should be “λmaxem”
- “NBD”, “ABD” – these acronyms should be defined.
- “UV (6 W)” – specify wavelength.
- Fig 3 caption: “c” should be “(c)”
- “mainly with” – extra space
- “stimuli-responsible” should be “stimuli-responsive”
- “N-Acetyl cysteine” should be “N-Acetyl cysteine”
- Ref 8: journal title italic.

- Thank you very much for kind corrections, and we corrected as you pointed out.

REVIEWERS' COMMENTS:

Reviewer #1 (Remarks to the Author):

The authors have addressed all issues I raised. This revised manuscript is now acceptable for publication in Communications Chemistry.

Reviewer #2 (Remarks to the Author):

The authors have answered the comments and made a good reversion.

Reviewer #3 (Remarks to the Author):

The authors have addressed the comments from my initial review in a comprehensive fashion. I have also read the comments from the other reviewers and these comments seem to have been addressed in an equally comprehensive fashion also.

Accordingly I am happy to recommend that the manuscript is now accepted for publication.

I compliment the authors on their excellent work.

Response to Reviewers

Dear reviewers,

We deeply appreciate reviewers' careful proofreading and appropriate suggestions. All reviewers accepted revised version without change. Therefore, there is no response to referees in this final version. Thank you again for the kind review.

Sincerely yours,

Kosuke Namba

Reviewer #1 (Remarks to the Author in original submission):

The authors have addressed all issues I raised. This revised manuscript is now acceptable for publication in Communications Chemistry.

Reviewer #2 (Remarks to the Author in revised version):

The authors have answered the comments and made a good reversion.

Reviewer #3 (Remarks to the Author in revised version):

The authors have addressed the comments from my initial review in a comprehensive fashion. I have also read the comments from the other reviewers and these comments seem to have been addressed in an equally comprehensive fashion also.

Accordingly I am happy to recommend that the manuscript is now accepted for publication.

I compliment the authors on their excellent work.